# Site-Specific O-glycosylation of SARS-CoV-2 Spike Protein and Its Impact on Immune and Autoimmune Responses

**DOI:** 10.3390/cells13020107

**Published:** 2024-01-05

**Authors:** Franz-Georg Hanisch

**Affiliations:** Center of Biochemistry, Medical Faculty, University of Cologne, Joseph-Stelzmann-Str. 52, 50931 Cologne, Germany; franz.hanisch@uni-koeln.de; Tel.: +49-221-478-4493

**Keywords:** SARS-CoV-2, COVID-19, spike protein, O-glycosylation, site-specific glycosylation, immune response, vaccine, carbohydrate-specific antibodies, autoimmunity, autoimmune diseases

## Abstract

The world-wide COVID-19 pandemic has promoted a series of alternative vaccination strategies aiming to elicit neutralizing adaptive immunity in the human host. However, restricted efficacies of these vaccines targeting epitopes on the spike (S) protein that is involved in primary viral entry were observed and putatively assigned to viral glycosylation as an effective escape mechanism. Besides the well-recognized N-glycan shield covering SARS-CoV-2 spike (S) proteins, immunization strategies may be hampered by heavy O-glycosylation and variable O-glycosites fluctuating depending on the organ sites of primary infection and those involved in immunization. A further complication associated with viral glycosylation arises from the development of autoimmune antibodies to self-carbohydrates, including O-linked blood group antigens, as structural parts of viral proteins. This outline already emphasizes the importance of viral glycosylation in general and, in particular, highlights the impact of the site-specific O-glycosylation of virions, since this modification is independent of sequons and varies strongly in dependence on cell-specific repertoires of peptidyl-N-acetylgalactosaminyltransferases with their varying site preferences and of glycan core-specific glycosyltransferases. This review summarizes the current knowledge on the viral O-glycosylation of the SARS-CoV-2 spike protein and its impact on virulence and immune modulation in the host.

## 1. Introduction

Viral infections with a world-wide spread and increased mortality, like those caused by influenza A strain H1N1 in 1918–1920 or by the severe acute respiratory syndrome corona virus-2 (SARS-CoV-2), represent a considerable threat to global human health. While the influenza pandemic was estimated to have cost from about 20 to 50 million human lives, making it one of the deadliest pandemics in human history, the global death toll after two years of the COVID-19 pandemic reached 5.5 million documented cases, a number with a high degree of uncertainty [1]. According to current WHO estimates, up to 650.000 people die every year of flu-related causes. But although COVID-19 is thought to have a substantially higher mortality compared to most flu epidemics, the actual death toll is significantly lower than expected. The reasons for these discrepancies can be seen in the improvement of medical care standards over the last century, including intensive care medicine, but also in protective measures based on epidemiological knowledge, which helped to control viral spread. Last but not least, the rapid availability of vector or RNA-based vaccines has made a strong difference to the flu pandemic. In December 2022, about two years after the emergence of the SARS-CoV-2 pandemic, more than 5.46 billion people world-wide had received at least one dose of a COVID-19 vaccine, which corresponds to more than 70 percent of the world population. However, the share of the population with full vaccination fluctuates between countries, since most African countries range below 35% whereas in other areas, the full vaccination state reaches levels from over 50% to more than 80% of the population.

SARS-CoV-2 presents, on its surface, several glycosylated proteins. Among these are the most exposed spike (S) protein, the envelope (E), the membrane (M), and ORF3a proteins, which are involved in host tissue recognition and penetration or later stages of the pathogenesis. In the focus of this narrative review, only those studies will be highlighted, which explicitly refer to the site-specific O-glycosylation of the S protein and its implications for SARS-CoV-2 infectivity and host immunity.

## 2. O-Linked Glycosylation of SARS-CoV-2 Spike Protein Reveals Host Cell-Specific Patterns

### 2.1. General Aspects of Mucin-Type O-glycosylation on Enveloped Viruses

#### 2.1.1. Site Specificity of O-glycosylation

Enveloped viruses are covered by a host-derived membrane that is modified by virally encoded glycoproteins. As viruses hijack the host’s cellular O-glycosylation machinery, the site specificity of N-acetylgalactosamine (GalNAc)-addition to serines or threonines of viral proteins are determined by the host cell-expressed polypeptidyl GalNAc-transferases (ppGalNAc-Ts). Twenty isoforms of these enzymes were described, which exhibit distinct but partially overlapping site specificities with respect to their peptidic substrates [2,3,4]. At this point, the aspect of tissue or organ-specific glycosylation comes into play. Mucin-type O-glycosylation is initiated by ppGalNAc-Ts, which act partly in a co-ordinated fashion by introducing GalNAc residues that serve as anchors for lectin- or chaperon-like functions of other members of this enzyme family and promote further GalNAc addition at adjacent sites in the peptide [5,6,7]. Correspondingly, the specific ppGalNAc-T repertoires of cells or tissues directly determine the site-specific O-glycosylation patterns of proteins. In accordance with this, analyses of virion surface proteins having passed a certain organ site vs. recombinant proteins expressed in a cellular model have revealed dramatically different O-glycosites and glycan structures. 

#### 2.1.2. Structural Variation of O-glycosylation

The structural variation of O-glycans is controlled by the tissue and cell-specific repertoires of the glycosyltransferases involved in the formation of a highly variable core structure of mucin-type O-glycans, of their extensions by more or less elongated polylactosamine-type chains, and of peripheral modifications by α-linked monosaccharides [8]. Up to eight variants of mucin-type core structures were reported so far, which represent di- to tetrasaccharides formed starting from core-GalNAcs by adding Gal(1-3) (core 1), Gal(1-3) and GlcNAc(1-6) (core 2), GlcNAc(1-3) (core 3), and GlcNAc(1-3) and GlcNAc(1-6) (core 4) as major structures (Figure 1). Minor core-types (core 5 to core 8) were only rarely reported in specific tissues or cells or could, in the case of core 6, represent degradation products. The major core-types exhibit an organ-characteristic expression pattern [9]. Early work from the 1980s showed that mucins produced by bronchial epithelia express not only the ubiquitous core 1 and core 2 structures (Figure 1) but also considerable amounts of the core 3 and core 4 structures [10]. Except for bronchi, the only other tissues expressing these core structures were the colon and salivary glands [9]. Also, peripheral structures of O-linked glycans vary with tissue expression (blood group antigens). Two characteristic examples can be found (1) in the deep gastric glands, which express organ-specific α1,4-GlcNAc-capped O-glycans on gastric MUC6 mucin [11]; and (2) in gastric but not duodenal Trefoil Factor Family 2 (TFF2), which carries LacdiNAc as a specific terminal modification of N-linked chains [12]. 

## 3. B-Cell Responses to the SARS-CoV-2 Spike Protein Are Restricted by Heavy O-glycosylation of the Target

Among various other functions, carbohydrate modifications of proteins are generally known to sterically shield protein epitopes that might be targeted by antibodies. The primary target of SARS-CoV-2 viruses, the spike protein, is characterized by heavy N-glycosylation covering most of the protein epitopes with up to 22 N-linked chains. The O-glycosylation of SARS-CoV-2 spike proteins is, meanwhile, also characterized but exhibits much greater variability with respect to the sites of modification. This variability can be associated with tissue- or cell-specific differences in the repertoires of ppGalNAc-Ts and also in the ways of how viral proteins are presented to the glycosylation machineries in the Golgi. Compared to viral protein glycosylation within infected cells of target organs, a greater variability of spike protein O-glycosites can be expected in RNA-based immunization due to overload artefacts produced by the glycosylation machineries of muscle and lymphoid cells at the site of injection. From work with recombinantly expressed glycoproteins, it is known that the overload stress in transfected cells can result in the formation of immature glycoforms with reduced site occupancies and an incomplete biosynthesis of glycans. Hence, the final product presented at cell surfaces to the immune system exhibits a much greater heterogeneity with respect to the sites modified and to the structures of O-linked glycans. The consequence will be a variable access to the protein backbone of the spike protein and a concomitant variation in antibody responses. 

### 3.1. Site Specificity of O-glycosylation on Recombinant vs. Virion-Expressed Spike Protein

Naturally glycosylated spike proteins on virions that have passed the tracheobronchial tract epithelia are characterized by a restriction to extremely small areas of immunologically accessible proteins at the receptor-binding site [13]. This extreme restriction results from heavy N- and O-glycosylation that covers most of the spike protein surface. In addition to 22 N-glycosylated sites’ [14] varying numbers of serine and threonine positions were found to be O-glycosylated. While occupancies of N-linked sites of the spike protein do not differ considerably between virions and the recombinantly expressed protein from HEK-293 cells [15], the number and location of O-linked sites varies dramatically (Table 1). In some reports referring to the analysis of the recombinant trimeric S protein [16,17], one site was reported as occupied. The latter study, by Shajahan et al., from 2023 [17] goes beyond previous reports, as it details O-glycosites and O-glycoprofiles not only for the Wuhan-Hu-1 wildtype (WT), but also for several variants of concern (VOCs) of SARS-CoV-2 (Alpha, Beta, Gamma, Delta, and Omicron). Exclusively, position T323 was found to be O-glycosylated, primarily with core 1 glycans, but the site occupancies of T323 varied from higher (WT) to lower rates on the VOCs. The aspect of varying site occupancies is generally not addressed in the referenced studies of Table 1. However, all studies agree on one point: T323 in the receptor-binding domain (RBD) represents the position with the highest site occupancy.

Accordingly, a more comprehensive O-glycosylation analysis of the SARS-CoV-2 spike protein became possible by using biomimetic enrichment strategies for glycopeptides, which revealed 27 O-glycopeptides with 18 unambiguous O-glycosites (Table 1), a finding that refers to the extremely low occupancies of O-glycosites [18]. Another study, based on recombinant insect and HEK293 cell-expressed S proteins, reported 25 O-glycosites (Table 1), most of which were located near N-glycosylation sequons and, preferentially, in proximity to those sequons with unoccupied N-sites [19]. In accordance with these results, a high energy collision-induced dissociation MS study, using the Byonic (version 3.6.0) software platform for data evaluation, even revealed 43 O-glycosites [20]. cells-13-00107-t001_Table 1Table 1Identified O-glycosites on viral and recombinant SARS-CoV-2 spike protein ^1^.Virion-Extracted S Protein
Recombinant S Protein
Infected Human Cells [15]Insect (H5) [20]HEK293 [18]HEK293 [19]HEK293 [17]HEK293 [16]**T22****T22**




T29




S31



S60






**T73****T73**




T76

S94





T95




**T114**
**T114**


S116



**T124****T124**



S151







T167



S221


**T236**

**T236**


**T284****T284**



T286




S297




T299



T306/7




**T323****T323****T323**
**T323****T323**
**S325****S325**



T333




S345




S477






T478



T547



T572




T573



T604/S605




**T618**
**T618**




T630




T632


**S659****S659**




**S673****S673**



**T676**
**T676**


**T678**
**T678**

T696






T716


T724





T732




T791




**S803****S803****S803**


**S810****S810**



**S813****S813**



T912




T939




S940




T941





S975



T1066



**T1076****T1076**
**T1076**

**T1077****T1077**
**T1077**

**S1097****S1097**
**S1097**

**T1100****T1100**




T1105





S1123




T1136



T1160




S1161




S1170




**S1175****S1175**



S1196



^1^ O-glycosites are located within the RBD domain (red frame), the NTD domain (blue frame), or the HR1 (grey) and HR2 domains (green). O-glycosite information was taken from references [15,16,17,18,19,20], as indicated in the table heading. Sites in boldface refer to those which were found by different groups to be O-glycosylated. Only the data reported in [15] refer to the O-glycosylation state of the virion-derived S protein from infected individuals (Wuhan-Hu-1 strain). The summarized data from reports [16,17,18,19,20] refer to the recombinant trimeric S protein expressed either in H5 insect cells or in HEK293 cells. Among these, the recently published paper by Shajahan et al. [17] gives insight into the O-glycosylation of several variants of concern (VOCs). Data on the recombinant monomeric S protein were not considered [19], as these revealed artificial over-glycosylation of the RBD domain.

Only one study [15] focused explicitly on the authentic O-glycosylation pattern of the extracted S protein from SARS-CoV-2 (WT) virions isolated from an infected individual (Table 1). In this study, the virion-expressed trimeric spike protein was found to be substituted at 17 O-glycosites [15], among which 14 sites were determined with diagnostic ions. The O-glycosylation patterns of the virion-expressed S protein was strikingly different from the recombinantly expressed glycoforms and occurred mainly in clusters within the S1 domain (11 sites), while the remaining six sites were restricted to the N-terminal region of the S2 domain. Again, a majority of these sites (11 out of 17) were located near N-glycosylation sites, which is in accordance with the findings by Bagdonaite et al. [19]. Of the 35 S/T sites located near N-glycosites, 11 sites were actually used by the O-glycosylation machinery. However, in contrast to the recombinant trimeric S protein substitution pattern, the occupied O-glycosites on the virion-extracted S protein cluster with occupied N-glycosites. Accordingly, the authors draw the conclusion that the dynamics of O- and N-glycosylation appears to be partially determined by an “O follows N” rule, which implies that the presence of N-glycosylated Asn is a prerequisite of O-glycosylation at S/T near N-glycosites [15]. However, there is no reason to assume that this rule is related to a lectin activity of some of the ppGalNAc-Ts, as the observed effects on transferase activities exhibit a certain degree of specificity for core-GalNAc-Ts [6]. Convincing support for the claimed rule can be seen in the O-glycosylation intensities at T618 in WT and the N616Q mutant S protein [15], since the latter remains non-glycosylated.

### 3.2. O-glycosylation and Its Impact on Infectivity

Looking at the O-glycosite patterns of the RBD domain, it is evident that the virion-expressed S protein exhibits a restriction to one O-glycosite (T323) [15]. Neither N331 nor N343 within the RBD domain were found to carry O-glycans adjacent to or within the sequon sequence at N+2 (T333 and T345). Contrasting with this restricted O-glycosylation, the recombinant insect cell-expressed glycoforms revealed up to five O-glycosites within the RBD domain (T323, S325, T333, S345, and S477) [20]. The latter study contrasts, however, with data from another report on an insect cell-expressed S protein, which did not confirm any of the previously described five O-glycosites within the RBD domain but found T478 glycosylated on both ectodomains, in close proximity to the receptor-binding motif which is important for interactions with ACE2 [19]. While information on the critical role of T478 glycosylation in viral receptor binding is lacking, a glycosylation of S494, one of the reported mutations within the RBD domain (D494S), has been claimed to enhance RBD-ACE2 interactions [21]. A predicted O-glycosylation of S686, which is located near the furin cleavage site, has also been discussed to have influence on the viral binding [17]. However, this site was never found to carry O-linked chains in any of the above cited studies (Table 1). In summary, it is evident that surface areas of the RBD domain, in proximity to the ACE2 interacting structures, remain unglycosylated to avoid steric hindrance. Accordingly, glycoproteomic data on recombinant monomeric RBD domains, as reported [19], are misleading, as they over-estimate the number of O-glycosites in this critical area (T315, T323, T333, T345, and T415). Moreover, the O-glycomic data reported for the recombinantly HEK-293-expressed monomeric S-RBD, which reveal the presence of sialylated or fucosylated core 2 hexasaccharides [22], may also merely reflect the generally observed O-glycome of the cellular expression system. 

The same holds principally true for site-specific O-glycan profiles obtained by MS/MS in higher-energy collisional dissociation (HCD) and electron-transfer dissociation (ETD) modes [18,20]. While the study by Zhang et al. refers to the HEK-293-expressed S1 subunit of the S protein [20], the paper by Dong et al. covers the entire S protein [18]. Both studies show strong variations in site-specific O-glycosylation and with respect to the structures of glycans identified at individual sites (Table 2). However, they agree on one major point: the analyses revealed S/T sites within the RBD domain to be heavily O-glycosylated, even with glycans of higher complexity that contain GlcNAc and form core 2, 3, or 4 structures. These RBD sites (refer to the red frame in Table 2) carry, similar to the majority of the other sites, primarily glycans belonging to core 1-based di- to tetrasaccharides. The only structural study referring to virion-expressed O-glycoprofiles [15] provides no evidence for more complex GlcNAc-containing O-glycans at T323 within the RBD domain. The O-linked sugars identified at this site were HexNAc and Hex-HexNAc, corresponding most likely to GalNAc and to the core 1 disaccharide. 

### 3.3. Implications of O-glycosylation for B-Cell Responses to the Spike Protein

The outlined current knowledge on the spike protein, and, in particular, the RBD domain-localized O-glycosites has strong implications for the design of SARS-CoV-2 vaccines, since only a quite narrow surface area of the spike protein RBD domain remains accessible to protein-directed immune responses. As referenced in the previous chapter, the virion-expressed spike protein isolated from an infected individual is characterized by a low density of glycosylation within the RBD domain. mRNA-based S protein vaccines, on the contrary, are undefined with respect to the O-glycosites and glycoprofiles produced at sites of immunization. The anti-SARS-CoV-2 vaccine mRNA containing LNPs are injected into the deltoid muscle and induce the production of spike protein in muscle tissues, the lymphatic system, and in the spleen. However, shedding of the spike protein or peptide fragments into the circulation and persistence of the immunogen for weeks are known to occur. These shed and fragmented versions of the spike protein could further add to the molecular heterogeneity of the immunogen and to the development of molecular mimicry effects. Moreover, as O-glycosylation is introduced independent of sequons into the protein, a variable and sub-stoichiometric site occupancy will expectedly result in extremely variable accessibilities of peptide epitopes on the spike protein surface. A cell-characteristic variation of O-glycosite patterns could explain, in part, the observed restricted efficacies of antibody repertoires and a facilitated escape of SARS-CoV-2 from immune surveillance. Restricted efficacies of B-cell responses may, however, counteract positive effects associated with broader repertoires of antibodies emerging in response to such heterogeneities in glycosylation. 

These considerations and assumptions were partly confirmed by a recent study, which compared the performance of a recombinant protein vaccine with that of a SARS-CoV-2 mRNA vaccine in single dose immunization [23]. The latter type of vaccine induced potent germinal center responses and elicited the production of neutralizing antibodies compared to the poor performance of the recombinant protein vaccine which exhibits, on average, higher degrees of carbohydrate shielding. It has been estimated that about 40% of the surface of the spike protein trimer is covered by glycosylation. 

## 4. Emergence of Carbohydrate-Specific Antibodies

### 4.1. General Aspects

A significant proportion of serum antibodies of the IgM and IgG classes recognize carbohydrate antigens. Most prominent are antibodies to terminally α-linked rhamnose, N-acetylglucosamine (GlcNAc), and N-glycolyl-neuraminic acid (NeuGc) [24]. Rhamnose and NeuGc are not regular components of human glycoconjugates, and this also holds true for other prominent targets of antibody responses in humans, like the two disaccharides Galα1-3Gal and GalNAcα1-3GalNAcβ. Antibodies to the former (the “Galili epitope”) can make up about 1% of circulating IgG, and those to the “Forssman antigen”, which is widespread on animal and bacterial cells, are also represented with high titers. Most importantly, the carbohydrate blood group antigens of the AB0, the Lewis and P systems, are defined by antibodies to xenoantigens. 

While the pathways of antigen processing and presentation are well characterized for peptides and proteins, little is known about the origin and maturation of carbohydrate-specific antibodies. Certainly, some of these antibody responses may be induced T cell-independently, but more recent evidence supports the assumption that, in particular, short glycan chains on glycopeptides induce the generation of antibodies via classical antigen presentation pathways and T cell-dependent activation. 

Although surface areas on proteins represent the predominant MHC-restricted epitopes for eliciting immune responses, a proportion of protein-linked glycans may be involved in binding to major histocompatibility complexes and in the recognition of glycopeptides by T-cell receptors [25]. Evidence for the potential of O-GlcNAc-modified peptides to bind to MHC class I and to induce a strong CTL response was provided by Harum et al. [26]. In 2002, we showed that dendritic cells (DCs) process MUC1 glycopeptides for presentation on MHC class II molecules without removing the carbohydrates [27]. Moreover, these DC-presented glycopeptides were demonstrated to be recognized by T cells, suggesting that a much broader repertoire of T cells can be expected against O-glycoproteins compared to responses based merely on their peptide sequences. More specifically, in the context of viral defense mechanisms, the HIV-1 envelope glycoprotein gp120 was recently shown to elicit a CD4+ T-cell repertoire that recognizes a glycopeptide epitope presented via the MHC class II pathway [28]. 

Even carbohydrates devoid of peptidic components can elicit B-cell responses mediated by classical MHC-restricted pathways. For example, some zwitterionic polysaccharides were shown to stimulate T cells by MHC class II-dependent interactions [29]. Later on, it was shown by Cobb et al. that zwitterionic polysaccharides are processed by a nitric oxide-mediated mechanism and are presented to T cells through the MHC class II endocytic pathway. Moreover, these oligosaccharides bind to MHC-II inside antigen-presenting cells (APCs) for presentation to T cells [30]. 

Of course, some carbohydrate antigens, including bacterial surface components, can activate B cells without T-cell help, since these type 1 antigens function as polyclonal B-cell activators. Another group of antigens (type 2 antigens) comprise polymers with repetitive motifs, such as polysaccharides of the GAG family.

### 4.2. Induction of (Auto)Immune Carbohydrate-Specific Antibodies in COVID-19 Patients

#### 4.2.1. Human Carbohydrate Antigens

Enveloped viruses hijack host cells and use their translation and glycosylation machineries, which results in the generation of viral protein cores with tolerogenic host-like glycan structures. However, these glycans are presented to the immune system in a foreign peptidic environment and hence can elicit immune responses that are directed either to carbohydrate-peptide mixed epitopes or to host-like glycotopes. It is not surprising, accordingly, that COVID-19 patients have unusually high titers of IgM and IgG antibodies directed to self-carbohydrates. According to a carbohydrate antigen microarray study, these cross-reactive self-carbohydrates comprise the glycolipid subgroup of gangliosides, protein N-linked glycans, protein O-linked lactosamine-type structures, and blood group-like structures [31]. 

Large IgG signals were reported for the group of gangliosides (asialo-GM1, GM1a, GD1a, and GD1b), which are frequently observed in patients with neurological disorders. However, reactivities to other members of the ganglioside family were also found, like GD3, fucosyl-GM1, Gb5, SSEA-4, and GM3. Anti-ganglioside antibodies exhibited more than tenfold higher titers compared to controls, a feature not observed in other viral infections, like HIV, and were not accompanied by antibodies to neutral glycolipids, which indicates a selective response [31]. The induction of these autoantibodies may be related to structurally similar glycotopes on the viral surface, in particular to cross-reactive carbohydrates on the spike protein. These cross-reactive structural elements are found, for example on GM3, asialo-GM1, GM1b, and GD1a which contain terminal NeuAcα2-3Galβ, Galβ1-3GalNAc, or NeuAcα2-3Galβ1-3GalNAc that are closely related to oligosaccharides in the group of mucin-type O-linked glycans (see below Section 5.2). 

Nearly all anti-carbohydrate IgM responses were lower in COVID-19 patients, indicating a general dysregulation of this antibody class [31]. The group of autoantibodies covers IgM and also IgG, which is reactive to certain truncated or immature (agalacto) complex-type N-glycans and to oligo-mannose structures found on high-mannose-type N-linked chains. Among reactivities to O-linked blood group-associated antigens, the sera of COVID-19 patients revealed significantly higher titers of antibodies to dimeric LacNAc (i-antigen), the blood group H1, and the sialyl Lewis x antigen.

#### 4.2.2. Non-Human Carbohydrate Antigens

On the contrary, abnormally low autoantibody titers to another glycolipid, the Forssman antigen (GalNAcα1-3GalNAcβ), and the iso-Forssman antigen, as well as to the structurally related core 5 (GalNAcα1-3GalNAcα–Ser/Thr) glycopeptide, correlated with a more severe disease. The Forssman antigen is known as a heterophilic antigen, which is not expressed in normal human individuals but has been reported to occur in transitional mucosae adjacent to carcinomas in colon tissues. The identification of this antigen is, however, based on antibody specificities, which might be cross-reactive between the Forssman antigen and the core 5 glycan. Also, another heterophilic antigen needs to be mentioned in this context, as it does not belong to the normal human antigen repertoire: the “Galili epitope”. Antibodies to this epitope were proposed to play a role in the evolutionary survival of primates in viral epidemics [32]. In line with this, a previous study had reported on an inverse relationship between COVID-19 disease severity and anti-α-Gal antibodies in patients’ sera [33]. Patients with the most severe outcomes had the lowest levels of anti-α-Gal antibodies. These antibodies represent, in normal individuals, up to 1% of the total IgG. In the study by Butler et al. [31], the α-Gal antigen displayed, however, inconsistent features with either higher or lower titers in different cohorts of COVID-19 patients relative to controls.

## 5. Disorders Associated with or Induced by Cross-Reactive Autoimmune Protein- and Carbohydrate-Specific Antibodies in COVID-19 Patients

### 5.1. Autoantibodies Induced by SARS-CoV-2 Infection Influence the Outcome of COVID-19

#### Autoantibody Responses to Protein Epitopes

The production of autoantibodies is a prerequisite and hence a key feature in the development and in the maintenance of autoimmune diseases. Although not fully understood, molecular mimicry is thought to form the basis of mechanisms that disturb the immunologic tolerance of host protein epitopes and elicit cross-reactive antibodies. Well-established examples of such antibodies are formed in response to the Epstein–Barr virus and systemically cross-react with a variety of host tissues causing widespread inflammation. 

As a possible complication of COVID-19, the cold agglutinin syndrome (CAS) and autoimmune hemolytic anemia have been reported [34]. These autoantibodies belong, generally, to the class of IgM and are characterized by their activities at lower temperatures and their reactivities against red blood cell antigens, including the carbohydrate blood group antigens. 

Protein arrays to measure IgG autoantibodies in sera of hospitalized COVID-19 patients revealed responses associated with connective tissue diseases, anti-cytokine antibodies, and anti-viral antibodies [35]. Autoantibodies were found in about 50% of patients, whereas less than 15% of the controls had detectable autoimmune responses. In a series of other studies, reviewed by Gao et al. [36], three groups of autoantibodies were found to be associated with severe cases of COVID-19: anti-nuclear antibodies (aNAs), anti-phospholipid antibodies (aPLs), and anti-type I IFN antibodies. The outcomes of aNA-positive patients were worse (death rate of 36.4%) than that of the autoantibody-negative patients (death rate of 13.6%). Anti-phospholipid antibodies can cause Antiphospholipid Syndrome, which is characterized by the rapid development of multiorgan thrombotic damage. The neutralizing IgG autoantibodies against type I IFNs, which develop in COVID-19 patients but not in asymptomatic or mild COVID-19 ones, were associated with severe disease courses, potentially due to their interference with the anti-viral effects of interferons. 

Rapid extracellular antigen profiling for autoantibodies against 2770 members of the exoproteome revealed a high prevalence of autoantibodies against immunomodulatory proteins, such as cytokines, chemokines, complement components, and cell-surface proteins [37]. Associated with this, the risk to develop diverse new-onset post COVID-19 autoimmune diseases within 3–15 months after infection is increased across all age groups and covers rheumatoid arthritis, systemic lupus erythematosus and vasculitis, as well as inflammatory bowel disease and type 1 diabetes mellitus [38]. Interestingly, when screening sera of vaccinated individuals (Pfizer–BioNTech mRNA vaccine BNT162b2), besides a production of neutralizing antibodies against the wild-type SARS-CoV-2, there is no obvious generation of IgG autoantibodies nor of anti-cytokine antibodies [39]. 

### 5.2. Carbohydrate-Directed Autoantibodies Are Associated with Autoimmune Diseases, Including Neurological Disorders

Alterations of host glycosylation patterns, as observed in patients suffering from autoimmune diseases, like Heymann nephritis, systemic lupus erythematosus, rheumatoid arthritis, mixed connective tissue disease, or scleroderma, are known to induce glycosylation-dependent autoantibodies [40]. Aside from this, many bacteria and enveloped viruses expose on their surfaces glycan structures, which are similar or even identical to those on host glycoproteins but are presented by proteins of the pathogen and hence in a foreign environment. 

Guillain–Barre syndrome (GBS) refers to an immune-mediated postinfectious syndrome which affects peripheral nerves and nerve roots in about one to two individuals out of 100.000 [41]. A variety of microorganisms have been associated with GBS, in particular, *Campylobacter jejuni*, the Zika virus, and, since about 2020, SARS-CoV-2. For all these, good evidence supports autoantibody-mediated processes that are triggered by molecular mimicry between surface epitopes of the pathogen and human peripheral nerve components, like gangliosides or sialoglycoproteins in myelinated axons [42]. Most patients develop neurological symptoms within four weeks post infection, associated with the following subtypes: acute inflammatory demyelinating polyradiculoneuropathy (AIDP) or acute motor axonal neuropathy (AMAN) [43]. A rarer subtype, the Miller–Fisher syndrome, presents ophtalmoplegia, ataxia, and areflexia as clinical symptoms. 

Autoimmune responses induced by SARS-CoV-2 and glycans on its spike protein show a cross-reactivity to gangliosides, a group of sialic acid-containing glycosphingolipids with a preferential expression in the central and peripheral nerve system, but it is also found as a regular constituent of lipid rafts in epithelial plasma membranes. Interestingly, there are a couple of structural elements in gangliosides of the GM1, GD1 series, which show strong similarities with elements of mucin-type O-linked oligosaccharides (Table 3). Most strikingly, the α-O-linked core 1 disaccharide, also known as the Thomsen–Friedenreich antigen, exhibits a structural relationship to the β-linked disaccharide forming a partial structure of GM1 and its asialo derivative. Cross-reactivities of monoclonal anti-TFα antibodies and the TFβ antigen are well known. Moreover, the respective sialylated core 1 trisaccharide forms a partial structure of GD1a. Both mucin-type O-glycans were described as structural components of the SARS-CoV-2 spike protein [18,20] and, unsurprisingly, asialo-GM1, GM1a, GD1a, and GD1b as the preferential autoantibody targets [31]. Anti-GD1b antibodies are closely associated with ataxia in GBS patients [44]. 

While the causal relationship between a SARS-CoV-2 infection and the potential development of carbohydrate-dependent autoimmunity in GBS can be regarded as supported by growing evidence, there is no such evidence for a relationship between vaccinations and the development of GBS [45]. But, also, the causal association between a SARS-CoV-2 infection and GBS is questionable. Systematic measurements of anti-ganglioside responses to GQ1b (derived from GD1b and GT1b), GM1, GD1a, and GD1b at a US reference clinical laboratory (January 2016 to March 2021) revealed that positivity rates for antibodies against the latter two had remained unchanged, while those against GQ1b and GM1 had even declined during the pandemic [46]. 

Since a group of S-type lectins, the β-galactoside-specific galectins, have been reported to play central roles in rheumatoid arthritis (RA) [47], there is reason to assume that this and related inflammatory autoimmune diseases, like polymyalgia rheumatica (PMR), could be related to anti-β-galactoside-specific antibodies emerging in response to virion-expressed O-glycans. In particular, galectin-1 and galectin-3, both of which recognize the O-linked core 1 glycan (TF antigen) could compete with autoantibodies to viral glycans. Evidence for this assumption in the context of the SARS-CoV-2 infection is, however, still lacking. Serum levels of galectin-1, galectin-3, and galectin-9 were, however, reported to be elevated in SARS-CoV-2 infected individuals and show a positive correlation with markers of inflammation [47], making them potential predictors of disease severity. Strikingly, the RBD of the SARS-CoV-2 spike protein exhibits considerable structural similarities to galectins (sequence and 3D structure), which might explain the capacity to bind to host carbohydrates and to facilitate, in this way, the viral entry into cells [48].

## 6. Conclusions

Looking from a global perspective, the roles of O-glycosylation in virus biology have been documented for a series of enveloped viruses, highlighting specific functions in various stages of viral infections, like the attachment to and entry into target cells, the assembly of viral proteins, and the exit of particles [49]. Although glycosylation of the SARS-CoV-2 spike protein covers about 40% of the protein surface, and studies on monomeric S-RBD suggest multiple O-glycosites with larger glycans [50], the RBD domain on the virion-expressed trimeric S protein does not get heavily O-glycosylated [15]. O-glycoproteomics from various groups have, meanwhile, revealed deep insights into the site-specific O-glycosylation of the S1 and S2 subdomains, which should enhance our understanding of the detailed structural requirements of virus binding to the ACE2 receptor. However, most of the studies do not report on authentic virus glycosylation, except for one study which refers to the S protein extracted from an infected individual [15]. Despite this, the studies agree on one point, namely, the recombinant trimeric S protein expressed in HEK293 cells exhibits only one site within the RBD domain (T323) with a considerable rate of O-glycosylation. This site carries short, unbranched chains of core 1 glycans with decreasing site occupancies when comparing SARS-CoV-2 WT and the more recently developed VOCs [17]. As the virus has evolved during the pandemic to higher transmissibility, it is reasonable to believe that a reduced O-glycosylation of T323 in the later-evolved strains contributes to increased infectivity.

The knowledge on S protein carbohydrate modifications certainly has a great impact on SARS-CoV-2 therapeutics associated with glycosylation and should support strategies for the development of more efficient vaccines. In this regard, not only the aspect of epitope masking or shielding of the protein surface should be considered in the context of viral escape mechanisms. Glycans expressed on the S protein also exhibit a certain degree of immunogenicity and could serve as targets in the anti-SARS-CoV-2 vaccination. The immunogenic potential of host O-glycans on the foreign viral protein surface is reflected in their ability to elicit undesired immune responses in infected individuals, which are known to cause (or are discussed to be associated with) autoimmune diseases. Future medical studies will have to identify and confirm the relationship of carbohydrate-mediated autoimmunity to post-COVID-19 diseases. A focus of such studies could be the demonstration of carbohydrate-dependent autoimmunity in the context of rheumatoid arthritis, where a central role of galectins in the initiation of inflammatory reactions is known and indicates that similar roles could be played by galactoside-specific autoantibodies. 

## Figures and Tables

**Figure 1 cells-13-00107-f001:**
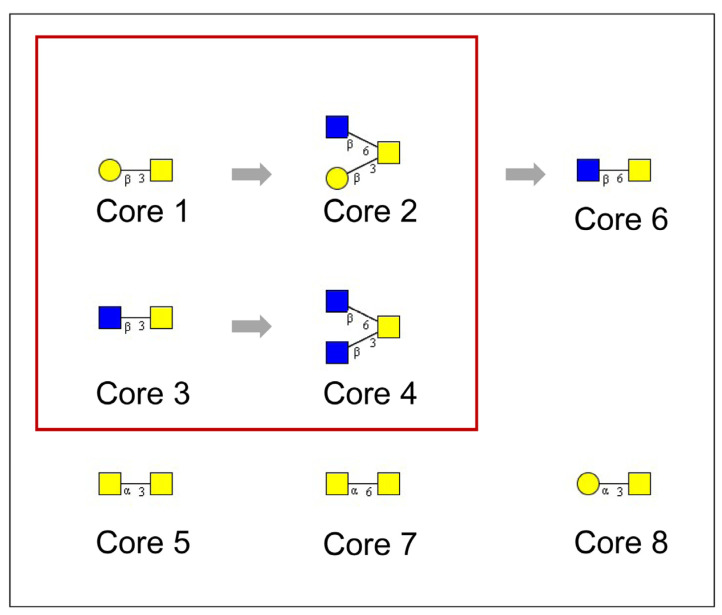
Core structures reported for mucin-type O-glycosylation of serine and threonine. Mucin-type O-glycans are formed by the stepwise addition of either D-galactose (yellow circle), N-acetyl-D-glucosamine (blue square), or N-acetyl-D-galactosamine (yellow square) to the core-GalNAc linked α-glycosidically to serine or threonine. Four major core types are formed (red frame), which are biosynthetically related (arrows) and show an organ-characteristic expression pattern. While core 1 and 2 exhibit a ubiquitous expression pattern, cores 3 and 4 were exclusively found on secreted mucins in the glandular epithelia of the bronchi, colon, and salivary glands. Cores 5 to 8 are less broadly and rarely expressed. Core 5, for example, was found to be restricted to mucins of meconium and intestinal adenocarcinoma.

**Table 2 cells-13-00107-t002:** Site-specific O-glycosylation profiles on recombinant SARS-CoV-2 spike proteins.

O-glycosite	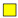	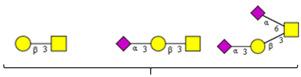 Core 1	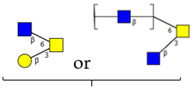 Core 2, Core 3, Core 4
T22		o	o	o	o
T29		o	o	o	o
S31		o	o	o	o
T33		o	o	o	
T73	+	+			+
T124	o				
S221					
T284	o			o	o
T286			o	o	o
S297	o	o	o	o	o
T299			o	o	o
T302			o	o	o
S305	o	o	o	o	
T307			o	o	o
T315			o	o	o
S316				o	o
T323	+o	+o	+	+	+
S325	o	o	+o	+o	+o
T547					
T572		o		o	o
T573		o	o	o	
T581		o		o	
T618					
T630				o	o
T632		o		o	
S637		o	o	o	o
T638		o	o	o	o
S640		o	o	o	
T645		o	o	o	
S659	o	o	o	o	o
S673	o	o	o	o	o
T676	o	o	o	o	o
T678	o	o	o	o	o
S680		o	o	o	o
T716					+
S803	+	+			+
S810	+	+	+	+	+
S813	+	+		+	+
S975					
S1123					
T1136					
S1175					+

o, site-specific O-glycosylation according to Zhang et al. [20]; +, site-specific O-glycosylation according to Dong et al. [18]. N-acetyl-D-glucosamine contained in glycans based on core 2 to core 4 of higher complexity. Red-framed O-glycosites are located within the RBD domain. The study by Dong et al. provides quantitative data for specific glycans at selected O-glycosites [18]. All sites (including T323) are occupied with glycans of defined composition at rates below 10%.

**Table 3 cells-13-00107-t003:** Structures of cross-reactive mucin-type O-glycan and ganglioside epitopes.

S Protein Epitopes	Ganglioside Epitopes
O-linked to Ser/Thr	
GM1a
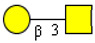	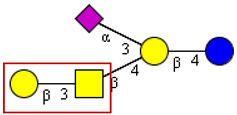
GM1b
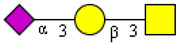	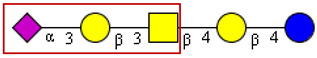
GD1a
	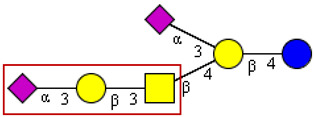

Blue circle, D-glucose; yellow circle, D-galactose; yellow square, N-acetly-D-galactosamine; pink rhombus, N-acetylneuraminic acid. The red frames highlight those structural elements of gangliosides with close similarity to mucin-type O-glycans. The structural information given in this figure does not refer to original data from other sources.

## Data Availability

Not applicable.

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
