# Peer review of "Site-Specific O-glycosylation of SARS-CoV-2 Spike Protein and Its Impact on Immune and Autoimmune Responses"

_cells, 2024, doi:10.3390/cells13020107_

Round 1

Reviewer 1 Report

Comments and Suggestions for Authors

This review summarises the current knowledge on O-glycosylation of the SARS-CoV-2 spike protein and discusses the impact on immune responses. Overall, the review is well written and most of the discussed topics are clear. However, with regard to the identified O-glycosylation sites on spike protein from different sources, some information is missing. It is important for the reader to have a comprehensive overview with all necessary details to better understand the functional role of O-glycosylation on viral glycoproteins.

In particular, the author should mention very clearly for every cited publication whether recombinant soluble spike, spike trimer or virion derived spike was analysed and mention the cells or tissues from where the proteins or virions were isolated (e.g. virions from infected cells or from patients). This is very important because the type of glycan is influenced by the cell type that produces the spike protein. A question related to this is whether any study reported so far is close to authentic virus glycosylation in an infected individual. The author should comment on that. Moreover, it should also be mentioned if all the presented data are from the original Wuhan variant or if there are also data from other SARS-CoV-2 variants. There are some recent studies on glycosylation of spike from different virus variants (e.g. Shajahan et al., 2023).

Several studies showed very low abundance of O-glycosylation sites – which challenges the relevance of O-glycosylation for SARS-CoV-2 spike function. This aspect should also be addressed more in detail in the review. Maybe the occupancy of the sites can be included in Table 1 or 2.

Minor points:

Table 1: HEK293 are also human cells, please replace human cells with infected human individual or something similar.

Table 1: why are some sites in bold?

Table 2: a line is missing in the symbol illustration for the core 2 glycan.

Table 2: why are there some brackets in the heading?

Table 2, line 314: change to “O-glycosylation”

Table 3, line 510: change to “N-acetyl-D-galactosamine”

Line 22: please remove “will try” from the sentence.

Line 30: thread should be “threat”

Lines 134-135: what does this mean “a maximum of two sites” – why was this pointed out here?

Line 139: does the low stoichiometry refer to low site occupancy?

Line 151: please add “region of” after N-terminal

Line 353: please provide references for the mentioned anti-carbohydrate antibodies.

Line 358: what does “highest antibody titers” mean in this context?

Line 523: GD1a or GM1b instead of GD1b?

Line 532: please check is GQ1b was explained before.

Line 603: the Watanabe et al., paper was published in 2020.

Author Response

Reviewer 2

In particular, the author should mention very clearly for every cited publication whether recombinant soluble spike, spike trimer or virion derived spike was analysed and mention the cells or tissues from where the proteins or virions were isolated (e.g. virions from infected cells or from patients). This is very important because the type of glycan is influenced by the cell type that produces the spike protein.

The respective information has been introduced into the revised version of the manuscript, if it was lacking. Actually, this issue had been addressed by the author, as it forms a central point for the assessment of authenticity of glycosylation. It is now explicitely mentioned in the text or in the legend to tables 1 and 2, whether the data refer to monomeric, trimeric, recombinant or virion-derived S protein. In case of studies on recombinant protein it is specified, in which cell type the spike protein was expressed. Data on recombinant monomeric S protein were not included in tables 1 or 2, as they are expectedly more far from an authentic O-glycosylation.

A question related to this is whether any study reported so far is close to authentic virus glycosylation in an infected individual. The author should comment on that.

It is mentioned several times in the text that the only report on SARS-CoV-2 spike protein, which is close to an authentic virus glycosylation, refers to the study by Tian et al. [15]. These authors had isolated virions from an infected individual (refer to supplementary data provided by authors) and extracted the spike protein prior to LC-MS on the (glyco)peptide level.  

Moreover, it should also be mentioned if all the presented data are from the original Wuhan variant or if there are also data from other SARS-CoV-2 variants. There are some recent studies on glycosylation of spike from different virus variants (e.g. Shajahan et al., 2023).

The author is grateful for this valuable information and included the study of Shajahan et al. from 2023, as it gives insight into glycosylation changes at T323 associated with evolution of SARS-CoV-2 from WT to a series of VOCs. Although the study is based on recombinant S protein (HEK293) and provides information only on one O-glycosite (T323), its significance lies in the finding that reduced site occupancies are associated with increased infectivities of the viral strains.

Several studies showed very low abundance of O-glycosylation sites – which challenges the relevance of O-glycosylation for SARS-CoV-2 spike function. This aspect should also be addressed more in detail in the review. Maybe the occupancy of the sites can be included in Table 1 or 2.

The author thank for this valuable suggestion. Unfortunately, most report do not report quantitative data with respect to specified O-glycosites or glycan structures located at these sites. The paper by Shajahan from 2023 refers to one site only (T323), and other papers reveal spotlight information on specific sites, like T618 in WT vs N616Q  mutant protein  [15]. The only paper reporting quantitative data on a series of O-glycosites and with specification of the glycan structures/compositions is the study by Dong et al [18]. As far as possible, the revised text and the legends to tables refer to this issue.

Minor points

All points listed under “Minor points” were considered during revision.

Reviewer 2 Report

Comments and Suggestions for Authors

The manuscript represents a comprehensive review of current knowledge about O-glycosylation of SARS-CoV-2 spike protein and relevant carbohydrate-specific antibodies in the context of COVID-19 disease. This review is well-written and will be beneficial to anyone who is interested in learning more about the complex glycobiological aspects of viral glycosylation and its relationship to carbohydrate-mediated autoimmunity to post-COVID-19 diseases. As a suggestion for minor revision, it would be worth to mention the role of endogenous lectins, e.g. galectins, as competitive partners of carbohydrate-specific antibodies (PMID: 8570910). Also, please introduce RBD abbreviation in the text.

Author Response

Reviewer 3

As a suggestion for minor revision, it would be worth to mention the role of endogenous lectins, e.g. galectins, as competitive partners of carbohydrate-specific antibodies (PMID: 8570910). Also, please introduce RBD abbreviation in the text.

The valuable suggestion of the reviewer was considered. In the revised text the role of endogenous lectins (galectins) in now mentioned. However, the author avoided to go into detail, as this aspect is only partially related to the reviewed topic. Please refer to the additional text in chapter 5.2.

RBD was defined in the text (please refer to line 143 in the revised text).

Reviewer 3 Report

Comments and Suggestions for Authors

This review paper by Hanisch O-glycosylation on Spike protein of SARS-CoV-2 is written in good format and shows the state-of-art for the related research field. This reviewer thinks this review is significant and should have a good readership. I recommend accept this article after minor revision.

1) English should be further polished.

2) The future direction and perspectives on the glycosylation (N/O and other types) should be expanded a bit to show the authors' own thinking.

Comments on the Quality of English Language

Minor->major proofreading is required.

Author Response

Reviewer 4

1) English should be further polished.

The text of the paper was read and partially revised by a native speaker to address the criticism.

2) The future direction and perspectives on the glycosylation (N/O and other types) should be expanded a bit to show the authors' own thinking.

Also the second point referring to the Conclusions was considered by introducing additional comments, which address the author’s thoughts on future research with relevance in the context of autoimmune diseases related to COVID-19.
